# Effect of Thermal Oxygen Aging Mode on Rheological Properties and Compatibility of Lignin-Modified Asphalt Binder by Dynamic Shear Rheometer

**DOI:** 10.3390/polym14173572

**Published:** 2022-08-30

**Authors:** Meng Cai, Xun Zhao, Xuanzhen Han, Peng Du, Yi Su, Cheng Cheng

**Affiliations:** 1School of Civil Engineering, Southwest Forestry University, Kunming 650224, China; 2Shuicheng Highway Administration of Guizhou Province, Shuicheng 553000, China; 3Yanlord Development (Tianjin) Co., Ltd., Tianjin 300090, China

**Keywords:** lignin, asphalt, short-term aging, long-term aging, rheological properties

## Abstract

Lignin is abundant in nature. The use of lignin in the asphalt pavement industry can improve pavement performance while effectively optimizing pavement construction costs. The purpose of this paper is to study the effect of lignin on the anti-aging properties of asphalt. Commercial lignin was selected to prepare a lignin-modified asphalt binder. The properties of lignin-modified asphalt were studied by rheological experiments. The high-temperature rheological properties of two kinds of base asphalt and modified asphalt samples with different contents of lignin under three conditions of original, rolling thin film oven (RTFO) aging, and pressure aging vessel (PAV) were tested and analyzed with temperature sweep, frequency sweep, and multiple stress creep recovery (MSCR) tests. By comparing the variation laws of evaluation indicators, such as complex shear modulus *G**, phase angle *δ*, anti-aging index, cumulative strain, and viscous component G_v_, we found that lignin could effectively improve the high-temperature stability of base asphalt, but it had a negative impact on the compatibility issues of base asphalt. Meanwhile, lignin played a filling role in the base asphalt, and the increase in viscosity was the fundamental reason for improving the high-temperature stability of the base asphalt. The research results indicated that lignin could effectively improve the anti-aging performance of asphalt and play a positive role in prolonging the service life of pavement.

## 1. Introduction

As a viscoelastic material, asphalt comprises a large proportion of high-grade road surfaces in many countries owing to its superior performance and driving comfort. However, asphalt has obvious shortcomings, such as high-temperature sensitivity and easy softening at high temperatures. It easily becomes brittle at low temperatures and may not meet the requirements of high-grade highways. The insufficient high-temperature stability caused by aging is also a concern for the road industry. The adhesion between asphalt and stone and the aging phenomenon of asphalt under the action of heat and oxygen affects the quality and durability of asphalt on roads [1,2,3,4,5]. In the process of high-temperature production, storage, transportation, and processing, asphalt comes easily into contact with oxygen in the air, resulting in short-term aging. The most important type of aging is thermo-oxidative aging. Thermo-oxidative aging can lead to changes in the chemical composition and molecular structure of asphalt, and temperature sensitivity may also change; moreover, different modified asphalts have different aging effects. Prolonged exposure to elevated temperatures also leads to an increase in viscosity of the asphalt binder and changes in its viscoelastic characteristics [6]. At present, styrene–butadiene–styrene block copolymer (SBS)-modified asphalt is widely used in China and other countries. SBS can significantly improve the high-temperature performance and anti-aging performance of asphalt and asphalt mixtures [7,8,9], but SBS-modified asphalt is expensive [10].

Lignin, as the second most abundant renewable natural polymer compound in nature after cellulose, is an aromatic polymer containing oxyphenylpropanol or its derivative structural units in its molecular structure [11]. Lignin can not only be used in phenolic resin materials [12,13,14], epoxy resin materials [15,16,17], and polyurethane materials [18,19,20] instead of raw petroleum chemical materials, but it can also be blended with polymer materials to increase the mechanical properties [21], thermal stability [22], anti-aging (oxidation) [23], and flame retardancy [24] of polymer materials. In recent years, it has been widely used in plastics [25], adhesives [26], and other fields. Wu et al. [27] analyzed the lignin-modified asphalt with infrared spectroscopy (FTIR) and differential scanning calorimetry (DSR) and pointed out that the addition of lignin can significantly delay the aging process of the asphalt and increase the thermal degradation of the lignin-modified asphalt after aging. Meanwhile, the stability and low-temperature crack resistance have been improved in the aging process. Gao et al. [28] studied the high-temperature rheological behavior and fatigue properties of lignin-modified asphalt binders. The results indicated that the addition of lignin improved the viscosity and the deformation resistance of asphalt under high temperatures at different rotating speeds. Batista et al. [29] the high-temperature, low temperature, and aging properties of a lignin-modified asphalt binder. The results demonstrated that lignin was conducive to improving the high-temperature rutting resistance and low-temperature crack resistance of the asphalt binder. Xu et al. [30] studied the rheological properties and anti-aging properties of a lignin-modified asphalt binder. The results showed that the addition of lignin helped to inhibit the formation of carbonyl functional groups in the asphalt binder after the RTFO and PAV aging processes. This indicated that lignin can be used as antioxidant modifier. However, the addition of lignin has an adverse effect on the fatigue resistance of the asphalt binder.

In conclusion, lignin can effectively improve the high-temperature stability and anti-aging performance of asphalt. However, the high-temperature rheological properties of lignin-modified asphalt under different aging effects should be further studied.

In this paper, the high-temperature rheological properties of two kinds of base asphalt and modified asphalt samples with different lignin contents under the three states of original, short-term aging, and long-term aging were analyzed by dynamic shear rheometer (DSR) temperature sweep, frequency sweep, and repeat creep test. Complex shear modulus *G**, phase angle *δ*, anti-aging index, cumulative strain, viscous component G_v,_ and other indicators were used to study the aging performance of lignin-modified asphalt.

The detailed plan completed in this paper is shown in Figure 1.

## 2. Materials and Methods

### 2.1. Raw Materials

#### 2.1.1. Lignin

The commercial lignin used herein was produced by Jinan Yanghai Chemical Co., Ltd. (Jinan, China). Lignin that passed a 200-mesh sieve was used for testing, and its main technical indicators, molecular weight, and pyrolysis parameters are shown in Table 1. Among them, the lignin indicator was provided by the manufacturer. The molecular weight of lignin was determined by Agilent pl-gpc50 gel chromatography, and the pyrolysis test was conducted with a Mettler Toledo tga/sdta851 synchronous thermogravimetric analyzer.

#### 2.1.2. Based Asphalt

In this paper, Maoming 70# asphalt and Donghai 90# asphalt were used for related experiments. The test results of various properties are shown in Table 2.

### 2.2. Preparation of Lignin-Modified Asphalt

The high-speed shear dispersion emulsifying machine (BME 100L) produced by Shanghai Weiyu Co., Ltd. (Shanghai, China), was used to prepare lignin-modified asphalt. The base asphalt was heated to 150 °C and kept for a period of time, and after passing through a 200-mesh sieve, the lignin was uniformly and slowly added to the base asphalt in batches during the low-speed shearing process for preliminary dispersion. Then, the speed of the high-speed shearing machine was gradually increased from low speed to 5000 r/min and then sheared for 1 h. After shearing, the sample container containing the mixture of lignin and matrix asphalt was put into a constant temperature oven at 120 °C for 1 h, and the preparation of lignin-modified asphalt was completed. In this experiment, five kinds of lignin-modified asphalts with different lignin contents (3, 6, 9, 12, and 15%) were prepared. The Maoming asphalts were marked as MM-3, MM-6, MM-9, MM-12, MM-15; Donghai Maoming asphalts were marked as DH-3, DH-6, DH-9, DH-12, DH-15; while the base asphalt was marked as MM-0 and DH-0. It was developed at 120 °C for 1 h and compared with the base asphalt to study the influence of its high-temperature rheological properties and aging properties.

### 2.3. Preparation of Aged Samples

According to JTGE 20-2011 “Highway Asphalt and Asphalt Mixture Test Regulations”, the rotating film oven test (T 0610-2011) and the pressure aging container accelerated asphalt aging test (T 0630-2011) were conducted. Firstly, 35 ± 0.5 g pitches were poured into glass bottles, the temperature of RTFOT was kept at 163 °C and the time was 85 min. The RTFO aged samples were prepared for subsequent testing. The sample plate with 50 ± 0.5 g of lignin-modified asphalt was put in the PAV long-term aging box, the temperature was set to 100 °C, the holding pressure was 2.1 MPa, and the aging time was 20 h.

### 2.4. Test Design and Evaluation Index

#### 2.4.1. Temperature Scanning and Frequency Sweep Test

The temperature sweep and frequency sweep tests of asphalt were carried out using a DHR-1 dynamic shear rheometer. The temperature sweep test was adopted through the strain control method with 12% of the target strain value and 10 rad/s of the loading frequency, and the test temperature range was 30~100 °C with 2 °C between the sampling intervals. The frequency sweep test was used to study the viscoelastic properties of the modified asphalt. The temperature of the frequency sweep was 30 and 60 °C, the frequency range was 0.1~100 rad/s, and the strain amplitude was 0.5%. Among them, the original asphalt and the asphalt after aging in the rotary film oven were scanned with a parallel plate with a diameter of 25 mm and a spacing of 1 mm, and the asphalt after pressure aging was scanned with a parallel plate with a diameter of 8 mm and a spacing of 2 mm.

#### 2.4.2. Aging Index Evaluation

The aging performance analysis based on high-temperature rheology was evaluated with the complex shear modulus aging index (*G^*^_AI_*), and the specific calculation formula is shown in Formula (1) [31]. The operating temperature range was 46~82 °C, and the sampling interval was 6 °C.
(1)$$\mathrm{Complex}\;\mathrm{shear}\;\mathrm{modulus}\;\mathrm{aging}\;\mathrm{index}\left(G_{AI}^{*}\right)=\frac{\mathrm{Short}\;\left(\mathrm{long}\right)\;\mathrm{term}\;\mathrm{aging}\;\mathrm{complex}\;\mathrm{shear}\;\mathrm{modulus}}{\mathrm{Unaged}\;\mathrm{complex}\;\mathrm{shear}\;\mathrm{modulus}}$$

#### 2.4.3. Repeat Creep Test

At present, the multi-stress repeated creep recovery test (MSCR) of AASHTO MP19-10 was adopted to evaluate the high-temperature performance of asphalt and modified asphalt [32]. In this paper, the multi-stress repeated creep recovery test was performed by loading for 1 s, unloading for 9 s, and 100 cycles of creep recovery process were conducted. The test temperature was 64 °C, and the test stress was 300 Pa. The viscosity component was obtained with the Burgers model. The Burgers model equation was divided into two equations; one was the stress–relaxation mode equation to input constant strain, the other was the creep loading mode equation with constant input stress. Both equations can be computed by inverse transformations and Laplace transforms. In this paper, the creep loading mode equation was adopted, and the Burgers fitting equation was formulated as (2):(2)$$\frac{\varepsilon \left(t\right)}{\sigma_{0}}=\frac{1}{E_{1}}+\frac{t}{\eta_{1}}+\frac{1}{E_{2}}\left(1-e^{-\frac{tE_{2}}{\eta_{2}}}\;\right)\;$$

Here, *ε(t)* is the cumulative creep strain of asphalt specimen; *σ*_0_ is the loading stress of the asphalt test; *η*_1_ and *E*_1_ represent the damping coefficient and elastic modulus in Maxwell’s model, respectively; *η*_2_ and *E*_2_ represent the damping coefficient and elastic modulus in the Kelvin model, respectively; t is the loading time; *E*_1_ reflects the elastic recovery ability of asphalt at high temperature; *η*_1_ is the viscosity coefficient reflecting the unrecoverable deformation, which is related to the viscosity deformation coefficient of asphalt; *E*_2_ and *η*_2_ reflect the load action under long-term load and at room temperature, which reflect the ability of asphalt to delay elastic recovery of deformation. In this study, *η*_1_ is the viscous part Gv of creep stiffness. The test was adopted to evaluate the high-temperature performance of base asphalt and lignin-modified asphalt.

## 3. Results and Discussion Results

### 3.1. Analysis of Rheological Properties of Lignin-Modified Asphalt

The test results of complex modulus *G** and phase angle *δ* for original and different aging asphalt samples are shown in Figure 2.

As seen in Figure 2a,b, compared with the original asphalt, the complex modulus of the modified asphalt increased owing to the addition of lignin. The lignin-modified asphalt showed the trend of reducing the complex shear modulus and increasing the phase angle with the increase in temperature, but the change trend gradually flattened. The increase in lignin improved the high-temperature performance, but did not change the temperature sensitivity, so temperature had an effect on it. As asphalt is a temperature-sensitive material, it exhibited elasticity when the temperature was low, and gradually transformed into a viscous flow state with the increase in temperature.

After comparing the effects of different lignin contents on the rheological parameters of the base asphalt, we found that for Maoming 70# base asphalt, the addition of too much lignin led predominantly to an increase in viscous resistance (internal friction). The increase in viscosity resulted from the decreased complex shear modulus and the increased phase angle. It was shown that the inflection point of the rheological parameters appeared when the lignin content was 9%, indicating that there was an optimal lignin content.

However, for Donghai 90# asphalt, the same trend did not appear, and the complex shear modulus and phase angle of the asphalt binder linearly increased and decreased with the increase in lignin content, respectively.

With the increase in lignin content, although the colloidal structure type of asphalt did not change, the increase in asphalt colloidal structure composition ration may have led to the change of its phase structure [33]. Compared with Donghai 90#, Maoming 70# had a higher proportion of asphaltenes [34]. Therefore, the asphaltene content was the fundamental reason that the different rheological parameters of the two base asphalts changed with the lignin content.

It can be seen in Figure 2c,d that the complex modulus and phase angle of each asphalt sample showed different degrees of decrease and increase after RTFO aging.

With the increase in lignin content, the complex modulus of Maoming asphalt first increased and then decreased, which was the same as that before aging. The phase angle was larger than that of original asphalt except for the content of 15%. Meanwhile, the complex modulus and phase angle of Donghai asphalt increased and decreased with the increase in lignin content, respectively.

In Figure 2e,f, the complex modulus of Maoming asphalt after PAV aging was opposite to that of the base and RTFO aging binder sample and decreased with the increase in lignin content, and the phase angle of Maoming lignin-modified asphalt was larger than that of the base asphalt. The complex modulus of Donghai lignin-modified asphalt was larger than that of the base asphalt except for the asphalt with lignin contents of 3 and 15%, which was consistent with the RTFO aging condition.

### 3.2. Analysis of PG Classification of Lignin-Modified Asphalt

The rutting factor *G**/sin*δ* was used to evaluate the rutting resistance of the asphalt binder at high temperatures. The larger the rutting factor *G**/sin*δ*, the better the high-temperature resistance and the stronger the permanent deformation resistance of the asphalt binder. Figure 3 shows the results for the rutting factor *G**/sin*δ* of each asphalt sample before and after aging at 58~82 °C.

As seen in Figure 3, with the increase in lignin content, the rutting factor *G**/sin*δ* value of the two kinds of asphalts before and after aging increased greatly, and the lignin-modified asphalt sample with 15% lignin content for Maoming asphalt showed the best anti-rutting performance after aging, but the PG high-temperature grade did not change. Donghai asphalt showed the strongest high-temperature rutting resistance when the lignin content was 15% before and after aging, and the high-temperature grade of PG increased by one level at this content.

### 3.3. Complex Shear Modulus Aging Index of Lignin-Modified Asphalt

In order to further clarify the aging degree, the complex shear modulus aging index *G***_AI_* in different test temperature ranges was calculated. The *G***_AI_* values of different binders under the RTFO and PAV aging conditions are shown in Figure 4.

It can be seen in Figure 4a,b that the *G***_AI_* of Maoming asphalt under RTFO aging decreased first and then increased with the increase in lignin content to values slightly less than that of base asphalt. The *G***_AI_* of the lignin-modified asphalt binder with 9% lignin content was the lowest. The RTFO aging index *G***_AI_* of the Donghai asphalt binder had no obvious regularity with the increase in lignin content, which may be the change in asphalt composition owing to aging. It can be seen in Figure 4c,d that the index *G***_AI_* under PAV aging changed in a parabolic form with the increase in temperature. The index *G***_AI_* of Maoming asphalt with the 9% lignin content under PAV aging condition was the lowest, which was consistent with the RTFO aging condition, indicating that the addition of lignin could effectively improve the resistance performance of Maoming asphalt under PAV aging conditions. However, the *G***_AI_* index of Donghai asphalt with 12% lignin content was the lowest, which was the same as the index under the RTFO aging condition. However, it did not mean that it had no aging resistance. The reason may be that this method could not be well characterized, and it further indicated that its applicability to asphalt with a high grade was not significant. To sum up, Maoming asphalt with a lignin content of 9% had the best anti-aging effect, and Donghai asphalt with lignin content of 12% had the best anti-aging effect.

### 3.4. Creep and Recovery Behavior of Lignin-Modified Asphalt

#### 3.4.1. Creep Test Viscous Component

Based on the repeated creep test, the viscosity component G_V_ value of creep stiffness was fitted by Formula (2) as the evaluation index of high-temperature stability performance [35]. The Burgers model was used to fit the curve at the creep loading stage to obtain the viscosity parameter, which was the viscosity part G_V_ of creep stiffness (as seen Figures 6 and 7). The value of Gv reflected the resistance of asphalt to permanent deformation. The larger the Gv, the better the rutting ability of the asphalt [36]. The unloading stage mainly reflected the measured viscous deformation and delayed elastic deformation. Figure 5 shows the creep recovery of asphalt with and without lignin for the first 10 cycles of the MM-0, MM-9, DH-0, and DH-12 samples at a stress of 300 Pa.

Under the same stress and temperature, the creep cycle data from the 1st to 100th times were fitted for two asphalts with different aging degrees, with an interval of 10 times. As seen in Figure 6 and Figure 7, the G_v_ values of the Donghai and Maoming binders had different changes in different aging degrees after adding lignin, which was long-term aging > short-term aging > before aging. This demonstrated that the addition of lignin can significantly improve the high-temperature resistance of asphalt. The *G**/sin*δ* and G_v_ values were consistent in evaluating the high-temperature performance of lignin-modified asphalt, but there were differences in the evaluation conclusions between different modified asphalts. However, the value of G_v_ suddenly increased during the PAV aging process of Donghai 90# asphalt after adding lignin, which may have been caused by the difference between the two matrix asphalt components.

#### 3.4.2. Accumulated Strain

As a typical viscoelastic material, asphalt has a certain delayed elasticity, and different kinds of asphalts and asphalts with different lignin contents had different recovery degrees. The delayed elasticity could be separated from permanent deformation by the creep recovery test. The initial strain in the recovery stage, i.e., the instantaneous unloading strain, was denoted by ε*_L_*. The residual strain at the end of the recovery stage was denoted by ε*_p_*, and ε*_L_*/ε*_p_* was used to denote the permanent deformation accounts for the total deformation, i.e., the proportion of the viscous part of the deformation. Selecting a single temperature level (64 °C), the ε*_L_*/ε*_p_* values of 10, 20, 30, 40, 50, 60, 70, 80, 90, and 100 loading times according to the abovementioned optimal content of asphalt before aging (MM-0, MM-9, DH-0, DH-12) were calculated. The results are shown in Figure 8.

It can be seen in Figure 8 that with the increase in the loading times, the ε*_L_*/ε*_p_* values of different lignin-modified asphalts also increased, which reflected the continuous accumulation of the permanent deformation of the asphalt with the increase in the loading times. The ε*_L_*/ε*_p_* values were very close, which was not similar to SBS-modified asphalt [37,38] and rubber-modified asphalt [39], indicating that the addition of lignin did not improve the elasticity of the asphalt and was only for filling. The reason may be that the molecular structure of lignin itself is complex. It has a three-dimensional network molecular structure, containing many aromatic groups and high carbon content [40], while the base binder is a mixture of extremely complex high-molecular hydrocarbon and non-metallic derivatives of hydrocarbons. Through the further analysis of the working mechanism in Figure 9, the addition of a lignin modifier led to the absorption of the asphalt liquid phase into the asphalt–lignin interaction area during the mixing process, forming an asphalt–lignin working system and changing the viscoelastic behavior of the asphalt binder [41].

The cumulative strain was reflected in the total residual strain of the asphalt sample after cyclic loading. The smaller the cumulative strain, the better the high-temperature resistance of the asphalt. In order to further illustrate that lignin had good high-temperature stability, the relationship between the cumulative strain and the number of cycle loadings is shown in Figure 10.

It can be seen in Figure 10 that the cumulative strain of asphalt increased with the increase in loading times, which was consistent with the actual load of the road. Under the same loading times, the cumulative strain of Donghai asphalt and Maoming asphalt before aging or RTFO aging decreased after adding lignin, indicating that the addition of lignin could reduce the temperature sensitivity of asphalt and give it better resistance to high-temperature deformation. It can be seen from the slope of the curve that the performance of each asphalt sample under RTFO aging tended to be consistent with that before aging, while in Figure 10c, it was found that the slope of the curve of MM-9 was significantly higher than that of MM-0, and the curve slope of DH-12 also had an increasing trend, indicating that the addition of lignin could effectively prevent the base asphalt from hardening after PAV aging, so as to prevent the aging of the base asphalt. Meanwhile, the slope of the MM-0 curve during PAV aging was greater than that of DH-0, indicating that the hardening degree of 70# asphalt was less significant than that of 90# asphalt after PAV aging. The main reason was that the content of heavy components in 90# asphalt increased during the aging process.

### 3.5. Compatibility Analysis of Lignin and Asphalt

Chang et al. [42,43] studied the rheological properties of compatible and incompatible polymer blends based on viscoelasticity theory and proposed a method to judge the compatibility of blends by taking the double logarithm curve of storage modulus (*G*′) and loss modulus (*G*″), also known as the Han curve. Using the Han curve to judge the compatibility of polymers, two basic conditions must be satisfied: (1) The *G*′-*G*″ logarithm curves at different temperatures are superimposed; (2) The slope of the curve at the low frequency end is equal to or close to 2. Through these two requirements, the compatibility between modifier and asphalt can be judged [44]. In order to further analyze the compatibility of the blends, the original binder and lignin-modified asphalt were analyzed with a van Gurp–Palmen (VGP) diagram [45]. The VGP diagram is a plot of the phase angle (*δ*) of asphalt against the corresponding complex shear modulus (*G**). The compatibility of the two asphalts and different contents of lignin was analyzed by frequency sweep of Han curve and VGP map at 30 and 60 °C, as shown in Figure 11.

It can be seen in Figure 11 that the Han curves of the two asphalts and the lignin-modified asphalt with different lignin contents were approximately straight lines at high temperatures before aging, and the slope of the Han curve was close to 2, which indicated that the asphalt binder belonged to a homogeneous mixed system at this temperature. The lignin had good compatibility with the matrix asphalt. However, the bifurcation phenomenon occurred in the unaged state at low temperatures, indicating that there was microscopic phase separation at low temperatures. After RTFOT aging, the two original asphalts and the lignin-modified asphalt with different lignin contents appeared to show bifurcation at a low temperature, but did not separate at a high temperature, indicating that the compatibility of original asphalt and lignin-modified asphalt was better in high-temperature conditions. After PAV aging, the base asphalt and modified asphalt both displayed the separation phenomenon at high temperatures, but there was no separation at low temperatures, which indicated that thermal oxygen and pressure aging promoted the decomposition of matrix asphalt and modified asphalt, resulting in large differences in internal molecular weight distribution. In the VGP curve, it was found that the aged Maoming 70# base asphalt and lignin-modified asphalt were superimposed at different temperatures, while the Donghai 90# base binder and lignin-modified asphalt were superimposed only before aging. The dispersion of different aging states could not be superimposed, indicating that the 70# binder was more compatible than the 90# binder.

## 4. Conclusions

In this paper, the aging properties of asphalt materials improved by a lignin modifier were evaluated in detail based on a rheological test. A series of tests were conducted on untreated and lignin-modified asphalt materials. Based on the test results, the following conclusions can be drawn:(1)The addition of lignin had a significant effect on the high-temperature resistance of asphalt, but the degree of improvement in the high-temperature performance of the two matrix asphalts was not the same. The results indicated that there was a compatibility problem with lignin in improving the performance of matrix asphalt.(2)The results of the repeated creep and recovery test indicated that lignin-modified asphalt and base asphalt showed the same behavior, and lignin did not increase the elastic recovery rate of modified polymer binders such as SBS. However, the addition of lignin increased the viscosity resistance of the asphalt binder, which significantly reduced the cumulative strain of the lignin-modified asphalt, and this was also the fundamental reason for improving the high-temperature stability of matrix asphalt.(3)After long-term aging, the cumulative strain of lignin-modified asphalt was higher than that of base asphalt, and the long-term aging performance was significantly improved. This was owing to the probable depolymerization and molecular weight reduction of lignin during long-term aging.

This study provided a new understanding of the aging properties of lignin-modified asphalt. Future research should focus on the thermal characteristics, field verification, and life cycle assessment of asphalt pavement with different lignin modifiers.

## Figures and Tables

**Figure 1 polymers-14-03572-f001:**
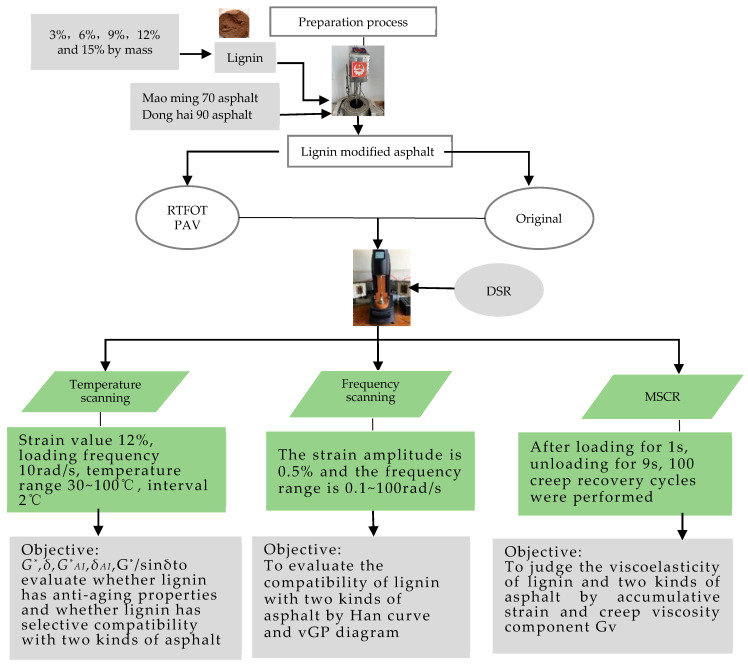
Experiment plan.

**Figure 2 polymers-14-03572-f002:**
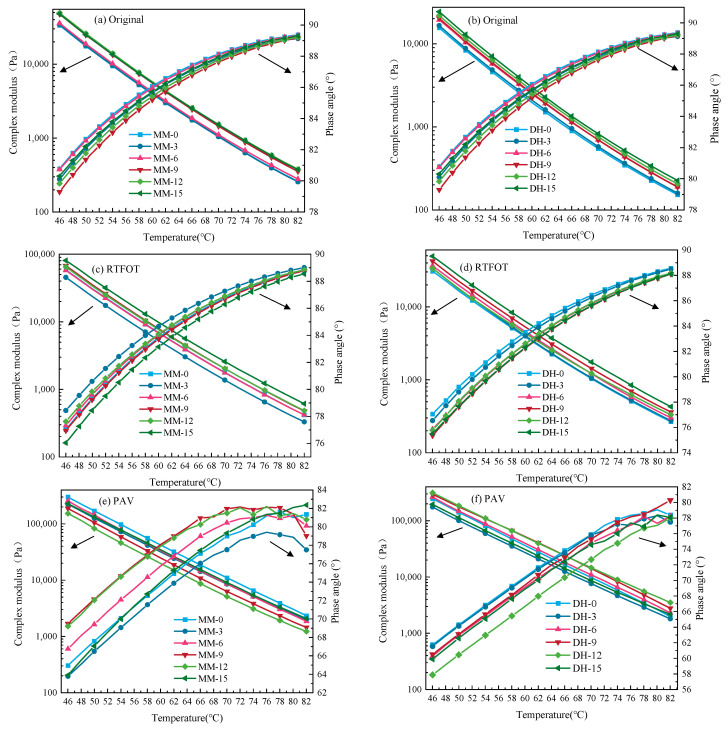
Results of complex shear modulus and phase angle of original and different aged asphalt with temperature. (**a**,**b**) represent the *G** and *δ* of different asphalts before aging, (**c**,**d**) represent the *G** and *δ* of different asphalts after short-term aging, (**e**,**f**) represent the *G** and *δ* of different asphalts after long-term aging.

**Figure 3 polymers-14-03572-f003:**
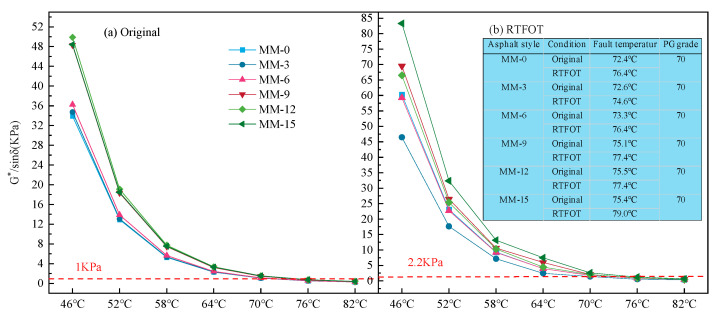
Rutting factor and PG grading of 70# and 90# asphalt with different lignin contents. (**a**–**d**) rutting factors representing different asphalts before and after short-term aging.

**Figure 4 polymers-14-03572-f004:**
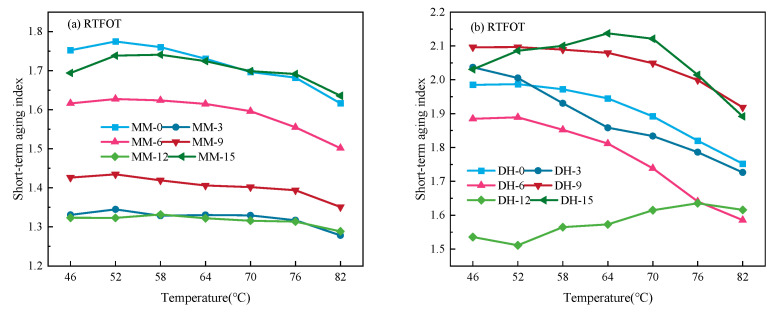
The results of *G*^*^_AI_ under different aging conditions. (**a**,**b**) represent the aging index of different asphalt with short-term aging, (**c**,**d**) represent the aging index of different asphalt with long-term aging.

**Figure 5 polymers-14-03572-f005:**
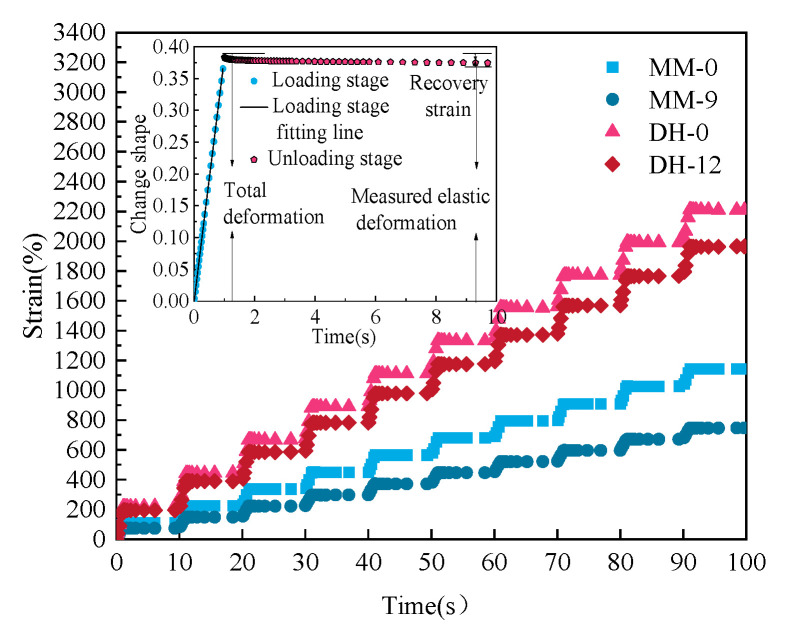
Burgers Model–Creep Test Diagram and Creep Recovery Diagram.

**Figure 6 polymers-14-03572-f006:**
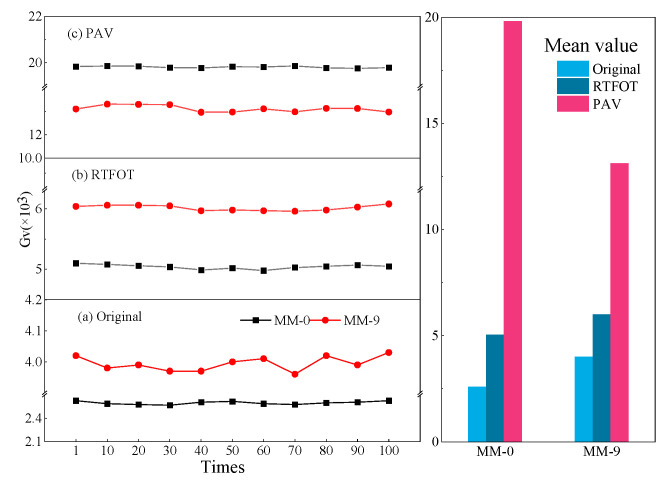
The viscous component G_v_ of the Maoming 70# binder and the lignin-modified binder. (**a**–**c**) creep test viscosity component representing as is, short-term aging and long-term aging.

**Figure 7 polymers-14-03572-f007:**
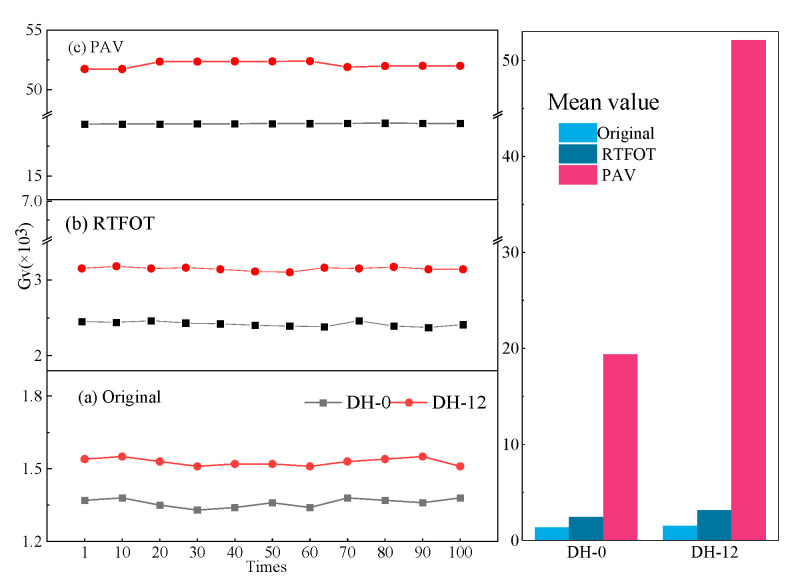
The viscous component G_v_ of the Donghai 90# binder and the lignin-modified binder. (**a**–**c**) creep test viscosity component representing as is, short-term aging and long-term aging.

**Figure 8 polymers-14-03572-f008:**
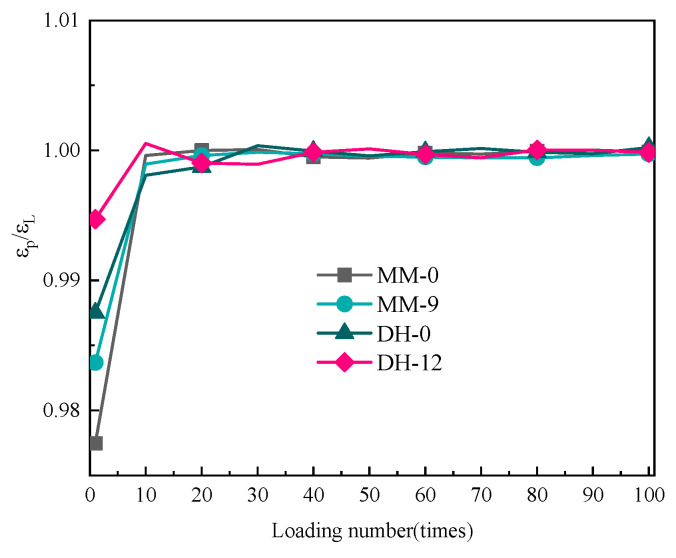
Results of ε*_L_*/ε*_p_* as a function of the number of loads before aging.

**Figure 9 polymers-14-03572-f009:**
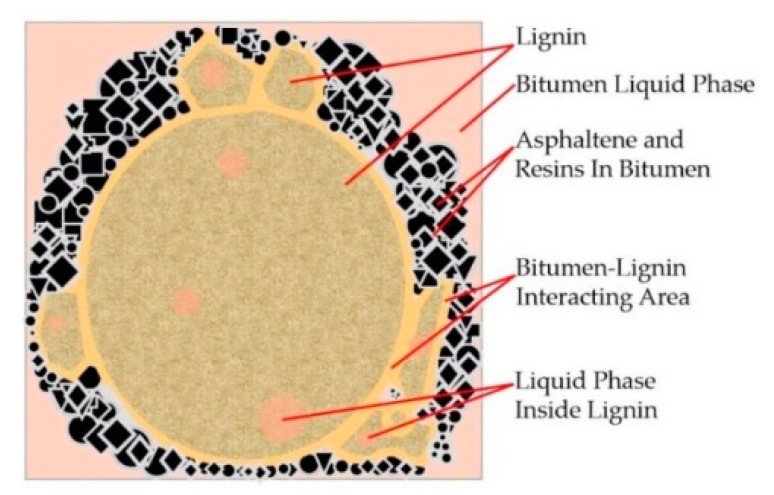
Schematic diagram of the mechanism of the asphalt–lignin working system [41].

**Figure 10 polymers-14-03572-f010:**
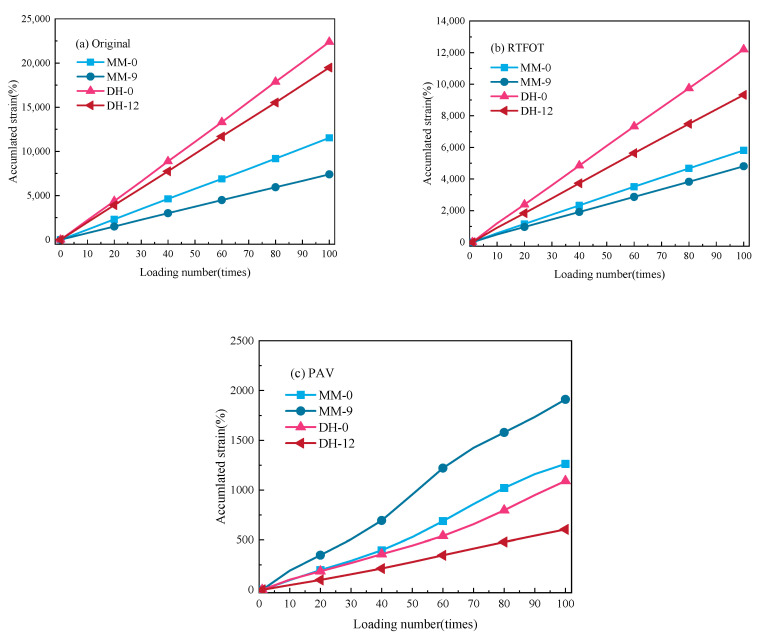
Relationship between cumulative strain and cycle loading times. (**a**–**c**) represents the cumulative strain of as is, short-term aging and long-term aging.

**Figure 11 polymers-14-03572-f011:**
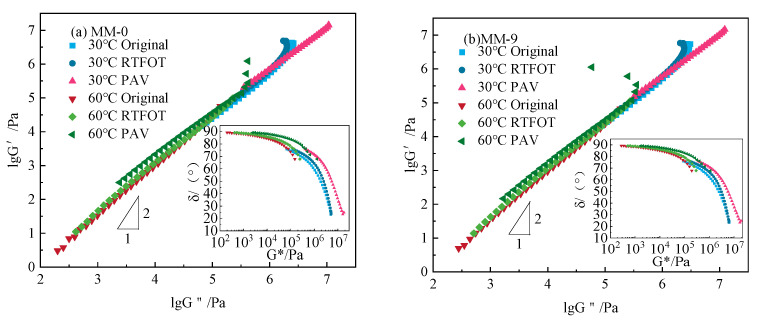
Han diagram and VGP diagram of two asphalts and lignin-modified asphalt under different aging conditions. (**a**–**d**) represent two types of asphalt with different temperatures and different aging.

**Table 1 polymers-14-03572-t001:** Main technical indexes, molecular weight, and pyrolysis parameters of lignin.

Figure	Main Specifications	Scope
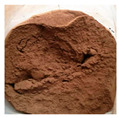	PH	7~8
	Appearance color	Brown powder
	Ash (%)	1
	Sugar content (%)	1~3
	Lignin content (%)	85~90
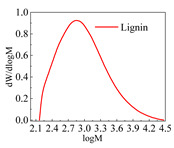	Number average relative molecular weight (M_n_)	960
	weight-average molecular weight (M_w_)	2964
	Peak molecular weight (M_p_)	1182
	Dispersion coefficient (d)	3.09
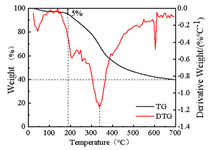	Initial decomposition temperature (°C)	189
	Maximum decomposition temperature (°C)	339.3
	700 °C Residue carbon rate (%)	39.7

**Table 2 polymers-14-03572-t002:** Maoming 70# and Donghai 90# base asphalt performance indexes.

Test Items	Unit	Test Result		Technical Requirement	
		Maoming 70#	Donghai 90#	70	90
Penetration (25 °C, 5 s, 100 g)	0.1 mm	65.4	86.2	60~80	80~100
Softening point	°C	46.6	45.7	≥46	≥45
Ductility (15 °C)	cm	>150	>150	≥100	≥100
TFOT penetration ratio (25 °C)	%	76.9	75.1	≥61	≥57
Residual ductility (15 °C)	cm	132.9	125.6	≥15	≥20

## Data Availability

Not applicable.

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
