# Peer review of "Effect of Thermal Oxygen Aging Mode on Rheological Properties and Compatibility of Lignin-Modified Asphalt Binder by Dynamic Shear Rheometer"

_polymers, 2022, doi:10.3390/polym14173572_

Round 1
Reviewer 1 Report
The article by Cai M. et al. examines the modification of asphalt binders with lignin. On the example of two grades of bitumen, the authors consider the effect of different amounts of lignin on their rheological properties, including after conducting artificial aging. This article requires numerous corrections before it can be accepted for publication in Polymers.
Specific comments on the article are as follows.
Title. The title of the article should not use an abbreviation, i.e., DSR should be written in its whole form or removed.
Line 31: “in my country” -> “in many countries”. There is no reason to highlight your country.
Line 33: Here and elsewhere, the authors use the past tense in respect to situations that take place also in the present moment. This is not correct. The authors should use the present tense in certain contexts. Here, the following replacement is needed: “It was easy to be brittle at low temperature and could not meet the requirements of high-grade highways, and the insufficient high-temperature stability caused by aging was also a problem that the road industry at home and abroad was concerned about” -> “It is easy to become brittle at low temperature and may not meet the requirements of high-grade highways, and the insufficient high-temperature stability caused by aging is also a problem that the road industry is concerned about”.
Line 38: “In the process of production” -> “In the process of high-temperature production”. All of the mentioned processes take place at elevated temperatures, which accelerate oxidation.
Line 39: “was easy to contact” -> “is easy to contact”.
Line 40: “one was thermo-oxidative aging” -> “one is thermo-oxidative aging"; “aging could lead” ->“aging can lead”.
Lines 40–42. Prolonged exposure to elevated temperatures also leads to an increase in viscosity of the asphalt binder and changes in its viscoelastic characteristics (see, for example, doi 10.1016/j.conbuildmat.2022.127946), which should also be noted.
Line 42: “wold also change” -> “may also change”; “asphalts had different” -> “asphalts have different”.
Line 43: “asphalt was used widely in China, it could significantly” -> “asphalt is used widely in China and other countries. SBS can significantly”. SBS-modified asphalt is used not only in China for enhancement of high temperature performance, see, e.g., doi 10.1134/S1061933X1404005X.
Line 45: “asphalt was expensive” -> “asphalt is expensive”.
Line 56: “lignin could significantly” -> “lignin can significantly”; “and increased the the thermal” -> “and increases the thermal”.
Line 58: “resistance had been improved” -> “resistance have been improved”.
Line 68: “It showed that lignin could be used” -> “This showed that lignin can be used”.
Line 69: “lignin had an adverse effect” -> “lignin has an adverse effect”.
Line 71: “lignin could effectively” -> “lignin can effectively”.
Line 73: “effects could be further studied” -> effects should be further studied”.
Line 76: The abbreviation DSR needs to be deciphered.
Line 80: “paper was shown” -> “paper is shown”.
Figure 1. This figure takes up too much space to be in the main part of the article. In addition, it carries no new information. This figure has to be transferred to the Supplementary Information.
Line 88: “weight were shown in Table 1” -> “weight are shown in Table 1”.
Table 1. All of the Main Specifications need to be specified in words along with the dimension units. The figures in the table are not needed; they can be given in the Supplementary Information.
Line 92: “properties were shown” -> “properties are shown”.
Tables 2 and 3: These tables should be combined into one.
Lines 105–107. It would be better if the asphalt labels contained the lignin concentration rather than the sample number. For example, DH-0 and DH-9 are asphalts containing 0% and 9% lignin, respectively.
Line 113: “were tested” -> “were conducted”.
Line 128: “Aging index evaluation index” -> “Aging index evaluation”.
Line 130: “formula was shown” -> “formula is shown”. Here, it is also necessary to specify at what angular frequency and temperature the complex modulus was measured.
Line 140: “Results and Discussion Results” -> “Results and Their Discussion”.
Line 142: “sample were shown” -> “sample are shown”.
Figure 2. Here, it is necessary to show with arrows which curves belong to the left axis (complex modulus) and which to the right axis (phase angle).
Figure 2d: “Complex aodulus” -> “Complex modulus”.
Line 150: "viscosity of the original asphalt" ->"complex modulus of the modified asphalt".
Line 153: “angle, but the change trend gradually flattened with the increase of temperature” -> “angle with the increase of temperature, but the change trend gradually flattened”.
Line 154: “asphalt was a temperature-sensitive material” -> “asphalt is a temperature-sensitive material”.
Line 159: “lignin led to an increase in viscous resistance” -> “lignin predominantly led to an increase in viscous resistance”.
Line 160: “showed the result that” -> “resulted from that”.
Line 164: “binder decreased linearly and increased linearly with” -> “binder linearly increased and decreased with”.
Line 167: What is the "gel ratio"? This information needs to be written or this term needs to be removed.
Lines 168–170: “Compared with Donghai … had a higher proportion of asphaltenes …with the lignin content.” The authors should either provide data on the content of asphaltenes in used asphalts or remove this speculation.
Line 174: “Maoming asphalt first increased”. Asphalt cannot increase or decrease. The authors must specify its variable characteristic – a complex modulus or something else.
Line 177: “asphalt decreased and increased with the increase” -> “asphalt increased and decreased with the increase”.
Line 180: “sample, which decreased” -> “sample and decreased”.
Line 200: “because the asphaltene content of the base asphalt binder changed at high temperature”. The asphaltene content in something does not change when the temperature gets higher. This phrase should be removed.
Line 205: “condition were shown” -> “condition are shown”.
Line 210: “content, which was all less than that of base asphalt” -> “content to values slightly less than that of base asphalt”.
Line 226: “the viscosity component GV value of creep stiffness was fitted…”. The authors should provide in the experimental part the methodology and equation for calculating the viscosity component GV.
Line 246: “caused by the change between” -> “caused by the difference between”.
Line 258: “denote that the permanent deformation accounts for the total deformation, which was the proportion” -> “denote the permanent deformation accounts for the total deformation, i.e. the proportion”.
Line 261: “to the above-mentioned optimal content of asphalt before aging (MM-1, MM-4, DH-1, DH-5)”. There was no mention above that these samples are optimal.
Line 262: “results were shown” -> “results are shown”.
Line 274: “lignin itself was complex, which was a kind” -> “lignin itself is complex and is a kind”.
Line 276: “binder was a mixture” -> “binder is a mixture”.
Line 286: “was shown” -> “is shown”.
Line 291: “It could be seen” -> “It can be seen”.
Line 295: “asphalt and had better resistance” -> “asphalt and make it better resistant”.
Line 296: “It could be seen from” -> “It can be seen from”.
Lines 312-313: “temperatures could be superimposed” -> “temperatures can be superimposed”; “end was equal” -> “end is equal”.
Line 316: “diagram was a plot” -> “diagram is a plot”.
Line 348: “results could be obtained” -> “results can be obtained”.
Line 354: “the same law, and did not increase the elastic recovery rate like other polymer modifiers such as SBS modified binder” -> “the same behavior, and lignin did not increase the elastic recovery rate of modified binder like other polymer modifiers such as SBS”.
Line 356: “increased the viscosity of asphalt binder” -> “increased the viscosity resistance of asphalt binder”. The authors did not measure the viscosity.
Line 357: “asphalt, which was also the fundamental” -> “asphalt, and this was also the fundamental”.
Line 361: “This was due to the depolymerization” -> “This was due to the probable depolymerization”. The authors did not measure the molecular weight of lignin after long-term aging.
Line 366: “This section is not mandatory but can be added to the manuscript if the discussion is unusually long or complex.” This should be removed.
Author Response
Responses to Reviewer Comments [polymers-1795305]
“Effect of Thermal oxygen aging mode on rheological properties and compatibility of lignin modified asphalt binder by Dynamic shear rheometer”
Reviewer #1
Comment #1: Title. The title of the article should not use an abbreviation, i.e., DSR should be written in its whole form or removed.
Author Response #1:
Thanks to reviewers for pointing it out, We have written the full name of DSR。
Lines 3-4, we added this text:
Dynamic shear rheometer
Comment #2: Line 31: “in my country” -> “in many countries”. There is no reason to highlight your country.
Line 33: Here and elsewhere, the authors use the past tense in respect to situations that take place also in the present moment. This is not correct. The authors should use the present tense in certain contexts. Here, the following replacement is needed: “It was easy to be brittle at low temperature and could not meet the requirements of high-grade highways, and the insufficient high-temperature stability caused by aging was also a problem that the road industry at home and abroad was concerned about” -> “It is easy to become brittle at low temperature and may not meet the requirements of high-grade highways, and the insufficient high-temperature stability caused by aging is also a problem that the road industry is concerned about”.
Line 38: “In the process of production” -> “In the process of high-temperature production”. All of the mentioned processes take place at elevated temperatures, which accelerate oxidation.
Line 39: “was easy to contact” -> “is easy to contact”.
Line 40: “one was thermo-oxidative aging” -> “one is thermo-oxidative aging"; “aging could lead” ->“aging can lead”.
Line 42: “wold also change” -> “may also change”; “asphalts had different” -> “asphalts have different”.
with polymer and solid nanosized additives[J]. Colloid Journal, 2014, 76(4): 425-434.
Line 45: “asphalt was expensive” -> “asphalt is expensive”.
Line 56: “lignin could significantly” -> “lignin can significantly”; “and increased the the thermal” -> “and increases the thermal”.
Line 58: “resistance had been improved” -> “resistance have been improved”.
Line 68: “It showed that lignin could be used” -> “This showed that lignin can be used”.
Line 69: “lignin had an adverse effect” -> “lignin has an adverse effect”.
Line 71: “lignin could effectively” -> “lignin can effectively”.
Line 73: “effects could be further studied” -> effects should be further studied”.
Author Response #2: Thanks for the reviewer's careful correction, We have completed the revision according to the reviewer's suggestions. See the red part of the article for details. In order to ensure the smooth language of the article, we are polishing it.
Comment #3: Lines 40–42. Prolonged exposure to elevated temperatures also leads to an increase in viscosity of the asphalt binder and changes in its viscoelastic characteristics (see, for example, doi 10.1016/j.conbuildmat.2022.127946), which should also be noted.
Line 43: “asphalt was used widely in China, it could significantly” -> “asphalt is used widely in China and other countries. SBS can significantly”. SBS-modified asphalt is used not only in China for enhancement of high temperature performance, see, e.g., doi 10.1134/S1061933X1404005X.
Author Response #3: We are very grateful and more than willing to accept reference suggestions from reviewers. The addition of these files can better provide readers with reference services.
The new reference is numbered [6]and [7]:
6.Yadykova A Y, Ilyin S O. Rheological and adhesive properties of nanocomposite bitumen binders based on hydrophilic or hydrophobic silica and modified with bio-oil[J]. Construction and Building Materials, 2022, 342: 127946.
7.Ilyin S O, Arinina M P, Mamulat Y S, et al. Rheological properties of road bitumens modified with polymer and solid nanosized additives[J]. Colloid Journal, 2014, 76(4): 425-434.
8.Zhang, W., Zou, L., Wang, Y. et al. Influence of High Viscosity Petroleum Resin (HV-PR) on the Intermediate and High Temperature Performances of Styrene–Butadiene–Styrene Block Copolymer (SBS) Modified Bitumen. Arab J Sci Eng (2022). https://doi.org/10.1007/s13369-021-06550-2
we revised this text:We added a sentence from you on lines 40-42, Prolonged exposure to elevated temperatures also leads to an increase in viscosity of the asphalt binder and changes in its viscoelastic characteristics
Comment #4: Line 80: “paper was shown” -> “paper is shown”. Line 88: “weight were shown in Table 1” -> “weight are shown in Table 1”. Line 113: “were tested” -> “were conducted”.
Line 128: “Aging index evaluation index” -> “Aging index evaluation”. Line 140: “Results and Discussion Results” -> “Results and Their Discussion”.
Line 142: “sample were shown” -> “sample are shown”. Line 142: “sample were shown” -> “sample are shown”.Figure 2. Here, it is necessary to show with arrows which curves belong to the left axis (complex modulus) and which to the right axis (phase angle).Figure 2d: “Complex aodulus” -> “Complex modulus”.
Line 150: "viscosity of the original asphalt" ->"complex modulus of the modified asphalt".
Line 153: “angle, but the change trend gradually flattened with the increase of temperature” -> “angle with the increase of temperature, but the change trend gradually flattened”.
Line 154: “asphalt was a temperature-sensitive material” -> “asphalt is a temperature-sensitive material”.
Line 159: “lignin led to an increase in viscous resistance” -> “lignin predominantly led to an increase in viscous resistance”.
Line 160: “showed the result that” -> “resulted from that”.
Line 164: “binder decreased linearly and increased linearly with” -> “binder linearly increased and decreased with”.
Line 177: “asphalt decreased and increased with the increase” -> “asphalt increased and decreased with the increase”.
Line 180: “sample, which decreased” -> “sample and decreased”.
Line 205: “condition were shown” -> “condition are shown”.
Line 210: “content, which was all less than that of base asphalt” -> “content to values slightly less than that of base asphalt”.
Line 246: “caused by the change between” -> “caused by the difference between”.
Line 258: “denote that the permanent deformation accounts for the total deformation, which was the proportion” -> “denote the permanent deformation accounts for the total deformation, i.e. the proportion”.
Line 262: “results were shown” -> “results are shown”.
Line 274: “lignin itself was complex, which was a kind” -> “lignin itself is complex and is a kind”.
Line 276: “binder was a mixture” -> “binder is a mixture”.
Line 286: “was shown” -> “is shown”.
Line 291: “It could be seen” -> “It can be seen”.
Line 295: “asphalt and had better resistance” -> “asphalt and make it better resistant”.
Line 296: “It could be seen from” -> “It can be seen from”.
Lines 312-313: “temperatures could be superimposed” -> “temperatures can be superimposed”; “end was equal” -> “end is equal”.
Line 316: “diagram was a plot” -> “diagram is a plot”.
Line 348: “results could be obtained” -> “results can be obtained”.
Line 356: “increased the viscosity of asphalt binder” -> “increased the viscosity resistance of asphalt binder”. The authors did not measure the viscosity.
Line 357: “asphalt, which was also the fundamental” -> “asphalt, and this was also the fundamental”.
Line 361: “This was due to the depolymerization” -> “This was due to the probable depolymerization”. The authors did not measure the molecular weight of lignin after long-term aging.
Line 354: “the same law, and did not increase the elastic recovery rate like other polymer modifiers such as SBS modified binder” -> “the same behavior, and lignin did not increase the elastic recovery rate of modified binder like other polymer modifiers such as SBS”.
Author Response #4: Thanks for the reviewer's careful correction, We have revised tenses and vocabulary according to the reviewers' suggestions. For more information, see the red section of the article
Comment #5: Line 76: The abbreviation DSR needs to be deciphered.
Author Response #5: Thanks to reviewers for pointing it out, We have written the full name of DSR。Dynamic shear rheometer
Comment #6: Line 92:Tables 2 and 3: These tables should be combined into one.
Author Response #6: We have fused the two tables together.
Lines 98, we added this text: Table 2 Maoming 70# (Donghai 90# )base asphalt performance index
|
Test items |
Unit |
Test result |
Technical requirement |
|||||
|
Maoming 70# |
Donghai 90# |
70 |
90 |
|||||
|
Penetration(25℃,5s,100g) |
0.1mm |
65.4 |
86.2 |
60~80 |
80~100 |
|||
|
Softening point |
℃ |
46.6 |
45.7 |
≥46 |
≥45 |
|||
|
Ductility(15℃) |
cm |
>150 |
>150 |
≥100 |
≥100 |
|||
|
TFOT penetration ratio(25℃) |
ï¼… |
76.9 |
75.1 |
≥61 |
≥57 |
|||
|
Residual ductility(15℃) |
cm |
132.9 |
125.6 |
≥15 |
≥20 |
|||
Comment #7:Lines 105–107. It would be better if the asphalt labels contained the lignin concentration rather than the sample number. For example, DH-0 and DH-9 are asphalts containing 0% and 9% lignin, respectively.
Author Response #7: We have mapped the lignin concentration of the asphalt label to the sample number. DH-0 and DH-9 are asphalts containing 0% and 9% lignin, respectively.
Lines 108, We modify it as follows:In this experiment, five kinds of lignin-modified asphalts with different lignin contents (3%, 6%, 9%, 12%, 15%) were prepared. The Maoming asphalts were marked as MM-3, MM-6, MM-9, MM-12, MM-15, Donghai Maoming asphalts were marked as DH-3, DH-6, DH-9, DH-12, DH-15, while the base asphalt was marked as MM-0 , DH-0 . It was developed at 120℃ for 1 h, and compared with the base asphalt to study the influence of its high temperature rheological properties and aging properties.
Comment #8: Line 130: “formula was shown” -> “formula is shown”. Here, it is also necessary to specify at what angular frequency and temperature the complex modulus was measured.
Author Response #8: Thanks for the reviewer's careful correction, Considering everything, we added a paragraph in line 137.The operating temperature range is 46~82°C, and the sampling interval is 6°C.
Comment #9: Line 167: What is the "gel ratio"? This information needs to be written or this term needs to be removed.
Author Response #9: What we want to say is Asphalt colloidal structure composition ratio.
Comment #10: Line 174: “Maoming asphalt first increased”. Asphalt cannot increase or decrease. The authors must specify its variable characteristic – a complex modulus or something else.
Author Response #10:
Lines 198, we revised this text: The complex modulus of Maoming asphalt first increased and then decreased, which was the same as that before aging. The phase angle was larger than that of original asphalt except for the content of 15%.
Comment #11: Line 200: “because the asphaltene content of the base asphalt binder changed at high temperature”. The asphaltene content in something does not change when the temperature gets higher. This phrase should be removed.
Author Response#11: Thanks to reviewers for pointing it out, Considering comprehensively, we have deleted this sentence.
Comment #12:Line 226: “the viscosity component GV value of creep stiffness was fitted…”. The authors should provide in the experimental part the methodology and equation for calculating the viscosity component GV.
Author Response#12: We introduced the formula from 145 to 160. The viscosity component was obtained by Burgers model, the Burgers model equation is divided into two equations,one is the stress-relaxation mode equation for input constant strain,the other is the creep loading mode equation with constant input stress, both equations can be computed by inverse transformations and Laplace transforms, in this paper, the creep loading mode equation is adopted, and the Burgers fitting equation is formulated as (2):
Type, ε(t) is the cumulative creep strain of asphalt specimen; σ0 is the loading stress of asphalt test; η1 and E1 represent the damping coefficient and elastic modulus in Maxwell's model respectively; η2 and E2 represent the damping coefficient and elastic modulus in the Kelvin model respectively; t is the loading time; E1 reflects the elastic recovery ability of asphalt at high temperature;η1 is the viscosity coefficient reflecting the unrecoverable deformation, which is related to the viscosity deformation coefficient of asphalt; E2 and η2 reflect the load action under long-term load and at room temperature, which reflects the ability of asphalt to delay elastic recovery of deformation. in this study, η1 is the viscous part Gv of creep stiffness.
Comment #13: Line 261: “to the above-mentioned optimal content of asphalt before aging (MM-1, MM-4, DH-1, DH-5)”. There was no mention above that these samples are optimal.
Author Response#13: We made a supplementary explanation in article 246 to 248. To sum up, Maoming asphalt with lignin content of 9% has the best anti-aging effect, and Donghai asphalt with lignin content of 12% has the best anti-aging effect.
Comment #14:Line 366: “This section is not mandatory but can be added to the manuscript if the discussion is unusually long or complex.” This should be removed.
Author Response#14: We sincerely thank the reviewers for pointing out formatting errors in the manuscript. We have deleted it.
Comment #15:Line 88: Table 1. All of the Main Specifications need to be specified in words along with the dimension units. The figures in the table are not needed; they can be given in the Supplementary Information.
Author Response#15: Thanks to reviewers for pointing it out, We have added units to the text
|
Feature |
Main Specifications |
Scope |
|
|
PH |
7~8 |
|
Appearance color |
Brown powder |
|
|
Ash (%) |
1 |
|
|
Sugar content(%) |
1~3 |
|
|
Lignin content(%) |
85~90 |
|
|
|
Number average relative molecular weight(Mn) |
960 |
|
weight-average molecular weight(Mw) |
2964 |
|
|
Peak molecular weight(Mp) |
1182 |
|
|
Dispersion coefficient(d) |
3.09 |
|
|
|
Initial decomposition temperature(℃) |
189 |
|
Maximum decompose temperature(℃) |
339.3 |
|
|
700℃Residue carbon rate (%) |
39.7 |
Comment #16: Lines 168–170: “Compared with Donghai … had a higher proportion of asphaltenes …with the lignin content.” The authors should either provide data on the content of asphaltenes in used asphalts or remove this speculation.
Author Response#16: Thanks for your comment. This conclusion is based on the research of reference of number 34( Chen,P.Q;Zhang,Y.Y.;Luo,Y,C.Discussion on Formation Mechanism of Epoxy Asphalt by Four-component Analysis. China Building Waterproofing, 2012, (10) :16-19. DOI: 10.15901/j.cnki.1007-497x.2012.10.004).

Reviewer 2 Report
In this work, the authors used two examples of asphalt for their studies. My question is why the Mao ming 70 and Dong hai 90 were selected and how the conclusions obtained from these two examples can be applied to other asphalt samples?
All the rheological measurements did not show an error bar in the plots.
Are there additional experimental evidence such as FTIR which can support the results from these rheological characterizations?
Author Response
Responses to Reviewer Comments [polymers-1795305]
“Effect of Thermal oxygen aging mode on rheological properties and compatibility of lignin modified asphalt binder by Dynamic shear rheometer”
Reviewer #2
Comment #1: My question is why the Mao ming 70 and Dong hai 90 were
selected and how the conclusions obtained from these two examples can be
applied to other asphalt samples?
Author Response #1:Thank you for your comment. These two kinds of asphalt are mainly used in China, of which 70 asphalt is used in South China and 90 asphalt is used in North China. Therefore, from the perspective of serving local projects. The conclusions need further verification to determine whether it can be applied to other asphalt.
Comment #2: All the rheological measurements did not show an error bar in the plots.Are there additional experimental evidence such as FTIR which can
support the results from these rheological characterizations?
Author Response #2: Thanks for your comment. There are many rheological experimental data, so the error bar is removed during data analysis. In fact, we conducted parallel experiments to ensure the accuracy of the experimental results.

Reviewer 3 Report
Manuscript ID: 795305
Title: Effect of Thermal oxygen aging mode on rheological properties and compatibility of lignin modified asphalt binder by DSR.
Reviewer comments:
1. Abstract is very lengthy and vague. Need to be shortened.
2. Fig. 1 is not necessary and this can be explained through text.
3. The data presented in table 1 is authors own experimental values? Justify.
4. Include the original images of the preparation of lignin modified asphalt.
5. Kindly explain Repeat creep test.
6. Section 3 Results and Discussion Results? What the authors are trying to explain fro this? There are several errors throughout the manuscript. Check thoroughly and correct them.
7. No visible difference in the graphs presented in Figs. 2 to 4.
8. Avoid point numbers in conclusion.
Author Response
Responses to Reviewer Comments [polymers-1795305]
“Effect of Thermal oxygen aging mode on rheological properties and compatibility of lignin modified asphalt binder by Dynamic shear rheometer”
Reviewer #3
Comment #1: Abstract is very lengthy and vague. Need to be shortened.
Author Response #1: Thanks for your comment. The abstract had been shortened.
Comment #2: Fig. 1 is not necessary and this can be explained through text.
Author Response #2: Thanks for your comment. Figure 1 is the outline of the full text, which can provide readers with direct writing information, and it is necessary to retain it.
Comment #3: The data presented in table 1 is authors own experimental values?
Justify.
Author Response #3: Authors got the data presented in table 1 by the tests. The indicators of lignin are provided by the manufacturer. The molecular weight of lignin is determined by Agilent pl-gpc50 gel chromatography, and the pyrolysis test is carried out by METTLER TOLEDO TGA / SDTA851 synchronous thermogravimetric analyzer
Comment #4: Include the original images of the preparation of lignin modified
asphalt.
Author Response #4: Thanks for your comment. The original images of preparation of lignin modified asphalt had been added.
lignin samples(origin RTFOT PAV)
DSR Shearing process
Comment #5: Kindly explain Repeat creep test.
Author Response #5: Thanks for your comment. The reason why rutting factor has limitations in the evaluation of high temperature performance of modified asphalt is that in addition to the test strain being controlled within the linear viscoelastic range of the material, it also includes: â‘ modified asphalt has a high delayed elastic recovery ability compared with ordinary asphalt; â‘¡ The loading mode of dynamic shear test is different from the load deformation response mode of pavement; â‘¢ There is a great difference between the reversible deformation mode of dynamic shear test and the cumulative development mode of pavement deformation.
The dynamic shear rheometer is used in the repeated creep test. The loading method is to load for 1s for creep test, then unload for 9s for deformation recovery, and complete 100 cycles of creep recovery process. This test method overcomes the defects existing in the G/sin test process. The test well simulates the deformation development process of the pavement under the action of driving load. At the same time, there is a deformation recovery period of 9s in the test process. It can consider the high delayed elasticity of the modified asphalt and comprehensively consider the high-temperature deformation resistance of the material. Therefore, the evaluation results obtained from the test can well characterize the high-temperature performance of the modified asphalt.
Comment #6: . Section 3 Results and Discussion Results? What the authors are trying to explain for this
Author Response #6: Thanks for your comment. According to various experimental data of lignin modified asphalt, the experimental results are explained, and the reasons for the experimental results of different asphalt and different aging are analyzed.

Reviewer 4 Report
1. Please add following information about the lignin
1.1. What was the source of the lignin, process and plant sources?
1.2. What is the Tg or softening point of lignin and its maximum temperature thermal stability. important information for any thermal process or lignin or other material
2. For samples name it is better to use MMc(no lignin), MM3, MM6, ....MM15 and the same for DH series. It will help to understand figure data directly.
3. Line 150-156 about FIg2 a-b is confusing. you mentioned" The lignin-modified asphalt showed the trend of reducing the complex shear modulus and increasing the phase angle, but the change trend gradually flattened with the increase of temperature" while it happened for control sample as well. it means the trend was not due to lignin.
4. All figures compared properties change vs temperature or time and it is hard to see the effect of lignin load on the properties. Understanding effect of lignin load is the main goal. due to logarithmic scale, it is hard to see the effect of lignin load on different properties. it is better to add figures with lignin load vs properties at least for the optimum condition.
5. Number of figures are too high (around 30 figures merged in 11 figures). It is better to reduce number of figures through consolidation of the results.
6. The article needs to prepare as a scientific article not a simple reporting of the results.
Author Response
Responses to Reviewer Comments [polymers-1795305]
“Effect of Thermal oxygen aging mode on rheological properties and compatibility of lignin modified asphalt binder by Dynamic shear rheometer”
Reviewer #2
Comment #1: What was the source of the lignin, process and plant sources?
Author Response #1: The commercial lignin used herein was produced by Jinan Yanghai Chemical Co., Ltd. (Shandong Province, China).
Lines 89, we added this text: The commercial lignin used herein was produced by Jinan Yanghai Chemical Co., Ltd. (Shandong Province, China).
Comment #2: What is the Tg or softening point of lignin and its maximum temperature thermal stability. important information for any thermal process or lignin or other material
Author Response #2: We added supplementary thermogravimetric test data to table 1
Lines 94, we added this text:
|
Feature |
Main Specifications |
Scope |
|
PH |
7~8 |
|
|
Appearance color |
Brown powder |
|
|
Ash (%) |
1 |
|
|
Sugar content(%) |
1~3 |
|
|
Lignin content(%) |
85~90 |
|
|
Number average relative molecular weight(Mn) |
960 |
|
|
weight-averag molecular weight(Mw) |
2964 |
|
|
Peak molecular weight(Mp) |
1182 |
|
|
Dispersion coefficient(d) |
3.09 |
|
|
Initial decomposition temperature(℃) |
189 |
|
|
Maximum decompose temperature(℃) |
339.3 |
|
|
700℃Residue carbon rate (%) |
39.7 |
Comment #3: For samples name it is better to use MMc(no lignin), MM3, MM6, ....MM15 and the same for DH series. It will help to understand figure data directly.
Author Response #3: We have mapped the lignin concentration of the asphalt label to the sample number. DH-0 and DH-9 are asphalts containing 0% and 9% lignin, respectively.
Comment #4: Line 150-156 about FIg2 a-b is confusing. you mentioned" The lignin-modified asphalt showed the trend of reducing the complex shear modulus and increasing the phase angle, but the change trend gradually flattened with the increase of temperature" while it happened for control sample as well. it means the trend was not due to lignin.
Author Response #4: We add that the increase of lignin improves the high temperature performance and does not change the temperature sensitivity, so temperature has an effect on it.
Lines 172-179, we revised this text: As could be seen from (a) and (b) in Fig. 2, compared with the original asphalt, the complex modulus of the modified asphalt increased due to the addition of lignin. The lignin-modified asphalt showed the trend of reducing the complex shear modulus and increasing the phase angle with the increase of temperature,but the change trend gradually flattened. The increase of lignin improves the high temperature performance, but does not change the temperature sensitivity, so temperature has an effect on it. As asphalt was a temperature-sensitive material, it exhibited elasticity when the temperature was low, and gradually transformed into a viscous flow state with the increase of temperature.
Comment #5: All figures compared properties change vs temperature or time and it is hard to see the effect of lignin load on the properties. Understanding effect of lignin load is the main goal. due to logarithmic scale, it is hard to see the effect of lignin load on different properties. it is better to add figures with lignin load vs properties at least for the optimum condition.
Author Response #4: Through comparison, we believe that logarithm can show more problems
Comment #6: Number of figures are too high (around 30 figures merged in 11 figures). It is better to reduce number of figures through consolidation of the results.
Author Response #6: We have made the diagrams together into a diagram. These diagrams can illustrate some problems
Comment #7: All figures compared properties change vs temperature or time and it is hard to see the effect of lignin load on the properties. Understanding effect of lignin load is the main goal. due to logarithmic scale, it is hard to see the effect of lignin load on different properties. it is better to add figures with lignin load vs properties at least for the optimum condition.
Author Response #7: Thanks for your comment. We mainly want to get the influence of lignin on asphalt performance. From the final result, we can achieve this result better. In the later research, we will fully consider your opinions and add more contrast charts to characterize the relevant performance.
Comment #8: The article needs to prepare as a scientific article not a simple reporting of the results.
Author Response #8: Thanks for your comment. The article had been revised so that it could be a scientific article not a simple reporting.

Round 2
Reviewer 1 Report
The authors have carefully corrected the article, and it can now be published in Polymers.
Reviewer 3 Report
The revised manuscript is improved and now I recommend this manuscript for publication in Polymers